# Cavus Foot Correction Using a Full Percutaneous Procedure: A Case Series

**DOI:** 10.3390/ijerph181910089

**Published:** 2021-09-25

**Authors:** Rodrigo Schroll Astolfi, José Victor de Vasconcelos Coelho, Henrique César Temóteo Ribeiro, Alexandre Leme Godoy dos Santos, José A. Dias Leite

**Affiliations:** Department of Surgery, Faculty of Medicine, Campus Porangabussu, Federal Universiy of Ceará, Fortaleza 60430-160, Brazil; jose_victor97@alu.ufc.br (J.V.d.V.C.); henriqueribeiro@ufc.br (H.C.T.R.); alexandrelemegodoy@gmail.com (A.L.G.d.S.); josealberto_leite@hotmail.com (J.A.D.L.)

**Keywords:** cavus foot, hindfoot osteotomy, minimally invasive

## Abstract

Cavus foot is a tri-planar deformity that requires correction in several bones and soft tissue. Minimally invasive surgeries are less aggressive, faster and easier to recover from. Here, we describe the initial results of a technique for percutaneous cavus foot correction. The procedure consists of calcaneal dorsal/lateral closing wedge osteotomy (with fixation), cuboid, medial cuneiform and first metatarsal closing wedge osteotomy (without fixation), and plantar fascia and tibialis posterior tenotomy with the patient in the prone position. Immediate weight bearing is permitted. Twenty patients were selected to undergo the procedure. The mean follow-up was 4.2 months and mean age 42.3 years. Eight of the 20 patients were submitted to cuboid and first metatarsal osteotomy, and 12 (60%) only calcaneal osteotomy. The median time for complete bone healing was 2.2 months. No wound complications were observed. No cases of non-consolidation of the cuboid or first metatarsal osteotomies were detected. The most common complication was sural nerve paresthesia. This is the first description of cavus foot correction using a minimally invasive technique. Complete bone healing is obtained even with immediate weight bearing and without cuboid and first metatarsal fixation.

## 1. Introduction

Cavus foot deformity is a common pathology in foot and ankle surgery [1] and although it has long been treated by orthopedic surgeons, since it is one of the most prominent deformities in poliomyelitis, numerous doubts and less-than-perfect surgical results persist [2,3]. Cavus foot repercussions include Achilles tendinosis, plantar pressure point formation and ankle instability [2,4,5].

The surgical treatment for cavus foot involves calcaneal valgization osteotomy, as described by Dwyer [6,7], dorsal wedge osteotomies [8,9,10], accessory procedures such as cuboid closing wedge osteotomy, first metatarsal or medial cuneiform dorsal closing wedge and soft tissue elongation [8,11,12]. Bone osteotomies are believed to be insufficient for severe cases and some professionals advocate the use of hindfoot arthrodesis even in healthy cartilage [13].

The open techniques are well described, and minimally invasive surgery (MIS) techniques have become increasingly popular in the last few years [10,11]. Some of the potential advantages of percutaneous techniques include faster procedures, because the soft tissue is left intact, low cost, fewer soft tissue complications, and less fixation, since weight-bearing naturally reduces longitudinal arch stiffness [10,11,14].

A number of percutaneous calcaneus osteotomies have been reported [10,11], but we were unable to find any articles describing three-dimensional deformity correction using percutaneous procedures. This article describes a technique for cavus foot MIS correction and presents some initial results.

## 2. Materials

Patients attended by to the Foot and Ankle Group of Federal University of Ceará were requested to participate in this study, twenty patients who subsequently met the inclusion criteria underwent minimally invasive surgery for cavus foot correction between July 2019 and February 2020. Inclusion criteria were Meary’s angle > 0 or calcaneal pitch angle > 30° in patients with pain caused by misaligned feet. All patients assigned the institutional informed consent as recommend by the Declaration of Helsinki.

Were considered pain due to the misalignment: pain on the lateral side of the foot, specially at cuboid and fifth metatarsal base; pain on the antero-lateral side of the ankle, pain on Achilles tendon insertion due to high calcaneal pitch and pain on the foot pressure points associated with callosity.

### Technique Description

The procedure is performed with the patient in the prone position with one sciatic nerve block. Two k-wires are placed (Figure 1A–C) percutaneously lateral to the bone so that the burr passage becomes limited by the k-wires, in the safe zone determined by Durston et al. [15], in order to determine the amount of bone that has to be removed for the lateral/dorsal closing wedge. Since x-ray images are essential, the surgeon must be aware that the perfect lateral view of the calcaneus is not the lateral view of the cavus foot.

A lateral portal is made, a 4.3 mm burr inserted (using a self-irrigated low-speed high-torque drill), and a windshield wiper movement performed between both k-wires to remove all the lateral, dorsal and plantar bone. The movement should be continuous to allow a more uniform amount of bone to be removed. More dorsal bone is removed if the cavus is due to high calcaneal pitch, and more lateral bone if the varus aspect is more important (Figure 2). A 20 mm/2.2 mm Shannon burr is inserted and the medial cortical bone transected. Screw fixation parallels open techniques and the reduction can be verified in both lateral and axial views.

Another portal is made under a fluoroscopic view, a lateral closing wedge performed at the center of the cuboid (Figure 3), and a punctual incision made lateral to the expected site of the superficial peroneal nerve. Next, the burr is inserted perpendicularly to the bone to avoid dissection or the passage of the burr through the soft tissue in this area, given that numerous unpredictable branches of the superficial peroneal nerve are present. Once the burr is inserted into the bone, the windshield wiper movement is repeated. Forced forefoot abduction completes the osteotomy, which does not usually include fixation.

With the patient still in the prone position, the first metatarsal base and/or medial cuneiform dorsal closing wedge can be performed with a 12/2.2 mm Shannon burr, always making an incomplete osteotomy, also without fixation. For this procedure, an additional regional block may be necessary.

A soft tissue procedure such as fasciotomy or flexor digitorum brevis tenotomy is performed at calcaneal insertion. In severe cases, tibialis posterior tenotomy is executed only at the navicular insertion, leaving the tibialis posterior spring ligament insertion intact. Weight-bearing is allowed immediately.

## 3. Results

Mean patient age and BMI were 42.3 years (SD 35.54–48.45) and 24.2 (SD 22.33–25.66), respectively. Mean follow-up time was 4.2 months (SD 3.4–4.5). Eight of the 20 patients (*n* = 20), (40%) underwent cuboid and first metatarsal osteotomy, and 12 (60%) only calcaneal osteotomy. The median time for complete bone healing was 2.2 months (SD 1.9–2.6). Mean hindfoot varus correction was 23.4° (SD 21.3°–25.6°). Twelve patients (60%) complained of sural nerve pain, 75% of whom recovered completely, the other three (25%) kept one small hypoesthesic rea on the nerve territory.

One patient with a Charcot sequela deformity evolved to delayed consolidation with screw fracture (Figure 4). Three cases (15%) evolved to incomplete correction, but only one (33.33%) required additional surgery. These incomplete correction cases belonged to the initial cases. No wound complications or non-consolidation of the cuboid bone or first metatarsal osteotomies were observed. (Figure 5).

Patients’ initial complaints resolved completely, as the median follow up time until this report was small (4.2 months) some unspecific complaints related to the rehabilitation program were still present especially muscular pain on long distance walks and non-painful limp.

## 4. Discussion

Cavus foot deformities occur in several planes and must be corrected in different bones in a multiplanar fashion. Traditional techniques are time-consuming and, in many cases, require changing the patient’s position on the operating table. Here, we presented a number of modifications to the usual descriptions of MIS, which allow the correction and good x-ray control in both views in only one position. The patient is moved from the transport trolley to the operating table in the prone position, where the sciatic block is made and the surgery performed immediately.

One of the principles of minimally invasive techniques is to preserve soft tissue since it can provide some stability to osteotomies and promote faster consolidation. Based on this principle, less fixation and immediate weight-bearing can be safely performed [14,15]. Immediate weight-bearing and the fact that the entire procedure can be performed with a regional block make it available for patients with severe comorbidities.

The safety of the lateral approach and the use of a Shannon burr has been well established [15]. The most widely used in dangerous structures are the medial calcaneal branch of the tibial nerve and posterior branches of the sural nerve. In cadaveric studies, although osteotomy crosses the nerve passages, soft tissue structures such as the quadratus plantaris protect them [15]. We believe that the high number of sural nerve complications, most of which are temporary, is due to overheating of the burr. Despite being irrigated throughout the procedure, it may still overheat because a large amount of bone must be removed from the same site.

The percutaneous approach requires a different set of surgical skills, and x-ray exposure is greater than in open techniques [15]. The learning curve was noteworthy, since the latest cases were all totally corrected, and procedure time declined from 1 hr 40 min to 40 min, reducing the x-ray exposition from 74 x-ray views to about 25 in the latest cases. Our first impression was that only non-severe cases could be addressed with MIS, but this is not the case. The most difficult cases are large feet with severe deformities. Making the closing wedge both lateral and dorsal allowed better correction in severe cases. Time to consolidation and amount of correction are similar to the ones commonly described for open techniques [16,17,18].

Cuboid osteotomies or opening wedge osteotomies of the medial cuneiform are important in cavus foot correction [1] because the forefoot adducts and rotates internally with the hindfoot varus [19,20]. Given that cuboid fractures usually evolve to shortening of the lateral column and that cuboid pseudarthrosis are not common, the percutaneous lateral closing wedge is expected to exhibit good consolidation [21]. The cuboid wedge is easily performed in the prone position and as expected, we had no problem with the eight cases that did not use fixation.

We prefer to do two incomplete dorsal closing wedge osteotomies with the thinner 12/2.2 mm Shannon burr in the first metatarsal and medial cuneiform rather than large wedges in only one bone, due to the lower risk for plantar cortical fracture and faster healing of these osteotomies.

Soft tissue procedures are carried out in most cases, with percutaneous release at the calcaneal insertion of the plantar fascia and flexor digitorum brevis, and in cases of more severe deformity, percutaneous tenotomy of the tibialis posterior only at the navicular insertion, maintaining the spring ligament insertion, in order to lengthen the tendon without totally transecting it. Soft tissue lengthening is commonly described as crucial to good correction [14].

This report has several limitations; we reported a small group of patients with a small follow up time and no standardized pre and post operatory quality of life score was applied.

## 5. Conclusions

To the best of our knowledge, this is the first description of successful MIS correction of cavus foot deformity and its initial results. Operating in the prone position made it possible to perform calcaneal, cuboid, medial cuneiform, and first metatarsal osteotomies and plantar and tibialis posterior tenotomies. The natural stretching of the plantar arch with weight-bearing makes the low fixation system of the minimally invasive procedure efficient. The most common complication is transitory sural nerve neuralgia.

## Figures and Tables

**Figure 1 ijerph-18-10089-f001:**
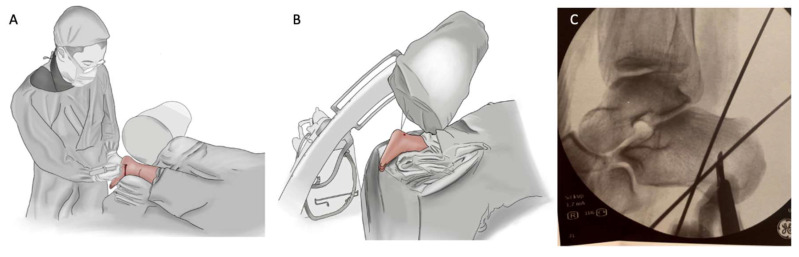
(**A**) The entire procedure can be performed with the patient in the prone position and C-arm position. (**B**) Two k-wires are positioned in a “v” shape, most of the proximal k-wire is placed at the limit of the “safe zone”, which is halfway between the calcaneal spur and fibula, and the position of the distal k-wire depends on the amount of correction needed. Note that the lateral calcaneal view is an oblique view of the ankle. (**C**) How k-wires are positioned in a X-ray view, showing how the "v" shape has to looks like in pratice.

**Figure 2 ijerph-18-10089-f002:**
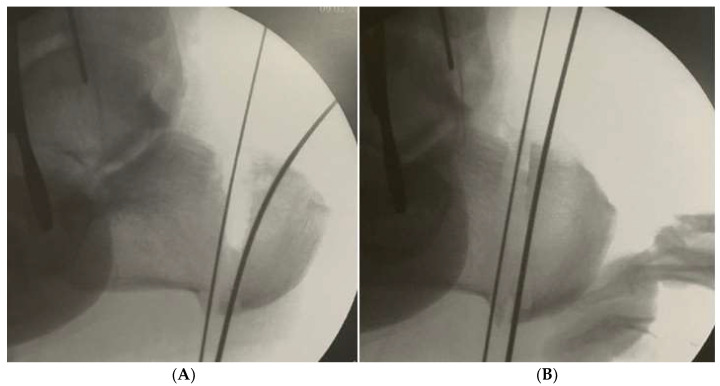
(**a**) Dorsal base closing wedge resected according to preoperative planning. (**B**) Dorsal base wedge closure.

**Figure 3 ijerph-18-10089-f003:**
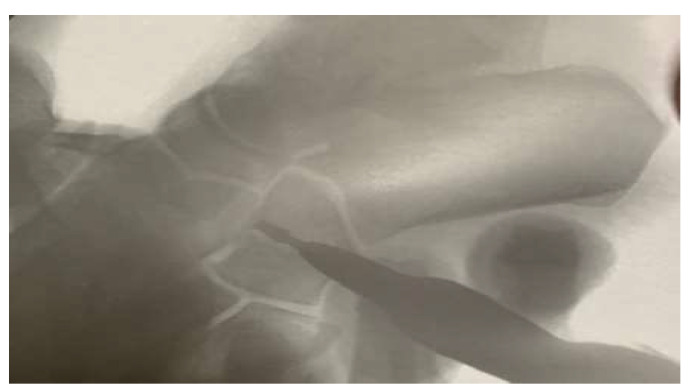
Cuboid view with the patient in the prone position.

**Figure 4 ijerph-18-10089-f004:**
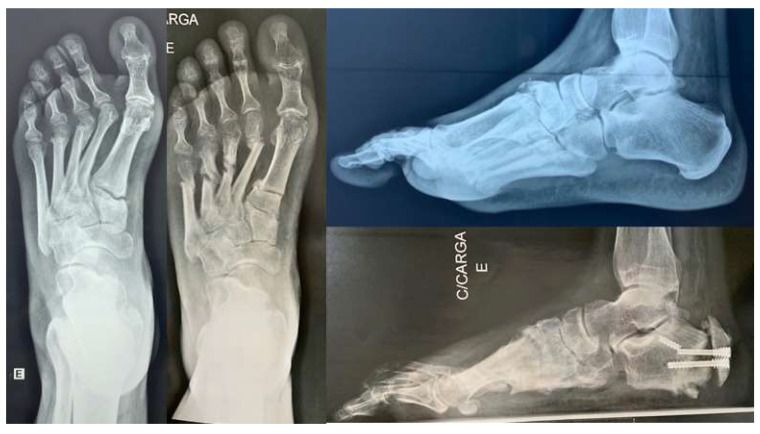
Delayed calcaneal consolidation in a Charcot sequela deformity.

**Figure 5 ijerph-18-10089-f005:**
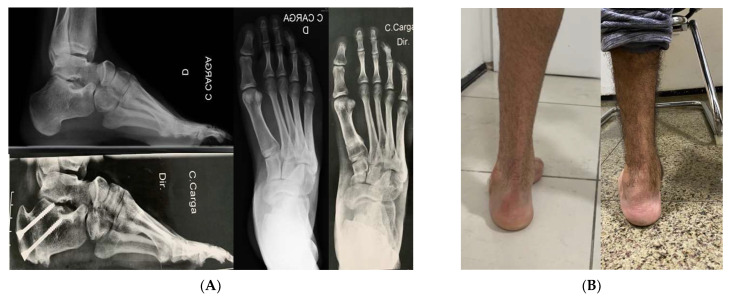
(**A**) We had full consolidation of cuboid and first metatarsal bones without fixation and with immediate weight bearing. (**B**) Clinical improvement of hindfoot alingment.

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
