# Peer review of "Cavus Foot Correction Using a Full Percutaneous Procedure: A Case Series"

_ijerph, 2021, doi:10.3390/ijerph181910089_

Round 1

Reviewer 1 Report

Dear authors:

It has been a pleasure to review your paper  “Cavus Foot Correction using a Full Percutaneous Procedure” and I have observed some errors that it’s necessary to change it. You can see below the recommendation

Title: It should include the type of study

Method

Where were the patients recruitment? Can you clarify this?

Can you include the SD of the age and the BMI?

 Results

You talk in the method section that the patients had pain. Can you explain anything the evolution of the pain in the results?

Discussion

Can you include a paragraph limitation of this paper and the conclusion.

I think that we don´t hold in this section about the pro and cons of this technique , can you improve this?

Author Response

Dear Reviewer,

Thank you for your important suggestions to our work, here we present the modifications we made.

Title: It should include the type of study

Thank you for your suggestion, we added the following at the title:

Cavus Foot Correction using a Full Percutaneous Procedure a Case Series

Where were the patients recruitment? Can you clarify this?

Thank you for your suggestion, we added the following at material section:

Patients attended by the Foot and Ankle Group of Federal University of Ceará were requested to participate to this study, twenty patients who subsequently met the inclusion criteria underwent minimally invasive surgery for cavus foot correction between 07/2019 and 02/2020. Inclusion criteria were Meary`s angle > 0 or calcaneal pitch angle > 30° in patients with pain caused by misaligned feet. All patients assigned the institutional informed consent as recommend by the Declaration of Helsinki.

Can you include the SD of the age and the BMI?

Thank you for your suggestion, we added the following at results sections:

Mean patient age and BMI were 42.3 years (SD 35.54 – 48.45) and 24.2 (SD 22.33 – 25.66), respectively. Mean follow-up time was 4.2 months (SD 3.4 – 4.5)

You talk in the method section that the patients had pain. Can you explain anything the evolution of the pain in the results?

Thank you for your suggestion, we added the following at the materials section:

Inclusion criteria were Meary`s angle > 0 or calcaneal pitch angle > 30° in patients with pain caused by misaligned feet Was considered pain due to the misalignment: pain on the lateral side of the foot, specially at cuboid and fifth metatarsal base; pain on the antero-lateral side of the ankle, pain on Achilles tendon insertion due to high calcaneal pitch and pain on the foot pressure points associated with callosity.

And we added the following at the results section:

Patient`s initial complains resolved completely, but as the median follow up time until this report was small (4.2 months) some unspecific complains related to the rehabilitation program were still present especially muscular pain on long distance walks and non-painful limp.

Can you include a paragraph limitation of this paper and the conclusion.

Thank you for your suggestion, we added the following at the materials section:

This report has several limitations; we reported a small group of patients with a small follow up time and no standardized pre and pos operatory quality of life score was applied.

Conclusion

To the best of our knowledge this is the first description of successful MIS correction of cavus foot deformity and its initial results. Operating in the prone position made it possible to perform calcaneal, cuboid, medial cuneiform, and first metatarsal osteotomies and plantar and tibialis posterior tenotomies. The natural stretching of the plantar arch with weight-bearing makes the low fixation system of the minimally invasive procedure efficient. The most common complication is transitory sural nerve neuralgia.

Reviewer 2 Report

It is a well planned case reports but there are some concerns:

1. Did patients sign an informes consent?

2. Add the ranges of all parameters expressed by means

3. Please, classify the cases using some of assessment parameters to report improvement comparing severity of previous deformaty/outcome achieved

4. Time of radiation

5. Clarify some sentence as "non-consolidation of the cuboid bone or first metatarsal osteotomies were observed"

6. Lack of comparisonof results  with other current techniques 

Author Response

Dear Reviewer,

Thank you for your important suggestions to our work, here we present the modifications we made.

  1. Did patients sign an informes consent?

Thank you for your suggestion, we added the following at the materials section:

All patients assigned the institutional informed consent as recommend by the Declaration of Helsinki.

  1. Add the ranges of all parameters expressed by means

Thank you for your suggestion, we added the following at the results section:

Mean patient age and BMI were 42.3 years (SD 35.54 – 48.45) and 24.2 (SD 22.33 – 25.66), respectively. Mean follow-up time was 4.2 months (SD 3.4 – 4.5)

  1. Please, classify the cases using some of assessment parameters to report improvement comparing severity of previous deformaty/outcome achieved

Thank you for your suggestion, we added the following at the results section:

Mean patient age and BMI were 42.3 years (SD 35.54 – 48.45) and 24.2 (SD 22.33 – 25.66), respectively. Mean follow-up time was 4.2 months (SD 3.4 – 4.5). Eight of the 20 patients (N=20), (40%) underwent cuboid and first metatarsal osteotomy, and 12 (60%) only calcaneal osteotomy. The median time for complete bone healing was 2.2 months (SD 1.9 - 2.6). Mean hindfoot varus correction was 23.40 (SD 21.30 – 25.60). Twelve patients (60%) complained of sural nerve pain, 75% of whom recovered completely, the other three (25%) kept one small hypoesthesic rea on the nerve territory.

  1. Time of radiation

Thank you for your suggestion, we added the following at the discussion section:

reducing the x-ray exposition from 74 x-ray views to about 25 in the latest cases

  1. Clarify some sentence as "non-consolidation of the cuboid bone or first metatarsal osteotomies were observed"

Thank you for your suggestion, we corrected the sentence:

No wound complications or non-consolidation of the cuboid bone or first metatarsal osteotomies were observed. (Fig. 5).

  1. Lack of comparisonof results  with other current techniques 

Thank you for your suggestion, we added the following at the discussion section:

Time to consolidation and amount of correction are similar to the ones commonly described for open techniques [19,20,21].

  1.  Weinheimer K, Campbell B, Roush EP, Lewis GS, Kunselman A, Aydogan U. Effects of variations in Dwyer               calcaneal osteotomy determined by three-dimensional printed patient-specific modeling. J Orthop Res. 2020 Dec;38(12):2619-2624. doi: 10.1002/jor.24772. Epub 2020 Jun 15. PMID: 32510162.
  2.  Moré L, Marchi Neto N, Ferreira RC. Efficacy of triple surgery for cavovarus foot in Charcot-Marie-Tooth disease. J Foot Ankle. 2021;15(1):19-25.
  3.  An TW, Michalski M, Jansson K, Pfeffer G. Comparison of Lateralizing Calcaneal Osteotomies for Varus Hindfoot Correction. Foot Ankle Int. 2018 Oct;39(10):1229-1236. doi: 10.1177/1071100718781572. Epub 2018 Jul 16. PMID: 30011380.

Round 2

Reviewer 1 Report

Accept in present form

Reviewer 2 Report

Thank you for your kind response.